## COMMENT

# Lessons from COVID-19 for behavioural and communication interventions to enhance vaccine uptake

Stephan Lewandowsky [1,2,3 ✉], Philipp Schmid [4,5,6],
Katrine Bach Habersaat [7], Siff Malue Nielsen[7], Holly Seale [8],
Cornelia Betsch [5,6], Robert Böhm [9,10,11], Mattis Geiger[5,6], Brett Craig [7],
Cass Sunstein [12], Sunita Sah [13], Noni E. MacDonald[14], Eve Dubé[15],
Daisy Fancourt [16], Heidi J. Larson [17,18], Cath Jackson[7],
Alyona Mazhnaya [19,20], Mohan Dutta [21], Konstantinos N. Fountoulakis [22],
Iago Kachkachishvili[23], Anna Soveri[24], Marta Caserotti [25],
Dorottya Őri [26,27], Giovanni de Girolamo [28],
Carmen Rodriguez-Blazquez [29], Maria Falcón [30], Maria Romay-Barja [31],
Maria João Forjaz [32], Sarah Earnshaw Blomquist[33], Emma Appelqvist[33],
Anna Temkina[34], Andreas Lieberoth [35], T. S. Harvey[36], Dawn Holford [1],
Angelo Fasce[37], Pierre Van Damme[38] & Margie Danchin[39,40,41]

The Covid pandemic has yielded new insights into psychological vaccine acceptance factors. This knowledge serves as a basis for behavioral and communication interventions that can increase vaccination readiness for other diseases.

Although the COVID-19 pandemic is widely considered to be over, vaccination remains the crucial tool to protect people from severe disease. Notwithstanding adequate supply, vaccine uptake varies considerably among countries and segments of society. For example, as of 30 June 2023, uptake of the primary course of vaccines in Europe ranged from 21.1% in Kyrgyzstan to 92.6% in Spain, and in the U.S. uptake is far higher among Democrats than Republicans with the gap exceeding 30% in some surveys. There were many reasons for low uptake, varying from country to country; however, a sizeable number of people across the globe chose not to get vaccinated. This hesitancy, much of it propelled by disinformation, has also spilled over into childhood vaccinations, with a notable decrease in confidence in 52 out of 55 countries polled by the United Nations International Children's Emergency Fund (UNICEF). Evidence-informed strategies for addressing low vaccine uptake are thus urgently required.

Focusing on those who make a decision not to vaccinate, we provide a toolbox of possible behavioural and communication interventions that are built on the recognition that vaccine hesitancy may arise from diverse psychological factors that require distinct interventions. We structure our interventions around the 7 C framework[1], which assesses vaccine hesitancy along the factors of confidence, complacency, constraints, calculation, collective responsibility,

A full list of author affiliations appears at the end of the paper.

**Table 1 Components of vaccination readiness according to the 7C model with suggested interventions for health authorities to improve vaccine acceptance.**

| Antecedents of vaccination readiness | Specific components of vaccination readiness | Recommended intervention | Links to resources for health authorities |
|---|---|---|---|
| Confidence | High trust in sender of information (e.g., community, faith and industry leaders; teachers and young people) | Identify vaccine champions or trusted members of the community to disseminate key information. | Identifying Opinion Leaders to Promote Behavior Change Empowering Community Leaders to Advocate for COVID-19 Vaccine |
| Trust in the security and effectiveness of vaccinations, the health authorities, and the health officials who recommend and develop vaccines. | High trust in health authorities and healthcare workers | Communicate clearly, transparently and with empathy with the public; use credible spokespeople, acknowledge uncertainty and do not over-reassure and hide negative information about vaccines. Diversify communication channels and platforms and prioritise key groups for communication. Equip Healthcare workers with knowledge, effective interpersonal vaccine communication skills, confidence and resources to recommend vaccines. Train health care workers to overcome mistrust and low confidence in vaccination among the public by debunking misinformation. | COVID-19 Vaccine Communication Handbook Communicating with Patients about COVID-19 Vaccination Communicating with Patients and the Public about COVID-19 Vaccine Safety Communicating with Health Workers about COVID- 19 Vaccination JITSUVAX Learning Resource to Debunk Vaccine Disinformation |
| | High trust in vaccine safety | Communicate vaccine safety surveillance mechanisms and both common and expected and rare but serious vaccine side effects. | Covid-19 Vaccines Safety Surveillance Manual |
| Complacency | Reasonable perceived risk of COVID-19 | Communicate disease severity, especially in at-risk groups. Encourage people to share narratives of severe disease. | Using Narrative Evidence to Convey Health Information |
| Complacency and competing priorities to get vaccinated due to low perceived risk of infectious diseases. | | Overcome myths around" super-immunity" in people who pursue perceived healthy lifestyles. | |
| Constraints Psychological hurdles in daily life that make vaccination difficult or costly. | Availability of trustworthy and transparent information | Improve access and awareness to trustworthy information. Strengthen community engagement and hold community meetings/forums to discuss vaccines and address concerns. Adapt scientific results to different cultural realities and make risk communication more accessible to marginalised and vulnerable groups. | COVID-19 Vaccines and Vaccination Explained WHO Outbreak Communication Guidelines Materials for Support Parenting under COVID-19 Considerations for Children and Adults with Disabilities How to Include Marginalized and Vulnerable People |
| | Availability of accessible information/awareness health literacy | Optimise translated resources. Make use of bilingual educators. Use different communication channels (e.g., hotlines, social media). | Cultural Adaptation of Health Communication Materials |
| | Low stress from the health crisis (e.g., job insecurity, political tension) | Manage concerns about health and stress and implement and communicate social security. | Coping with Stress |
| Calculation | High knowledge about COVID-19 disease risk and vaccine risk/benefit | Increase vaccine decision-making with measures such as decision aids, icon arrays, short videos etc. | Should I Get the COVID-19 Vaccine for my Child? Short Videos that Respond to Vaccine Questions Chatbot to address people's questions about Covid-19 |
| Degree to which personal costs and benefits are weighted. | High perceptions of vaccine benefits | Communicate personal benefits by highlighting vaccine effectiveness and safety profile. | How to Tailor COVID-19 Vaccine Information to Your Specific Audience |
| | Availability of personalised information | Provide information that is personalized to people's underlying medical and social histories. Share messages from | Research on Information Needs of Vaccine-Hesitant Adults |

**Table 1 (continued)**

| Antecedents of vaccination readiness | Specific components of vaccination readiness | Recommended intervention | Links to resources for health authorities |
|---|---|---|---|
| | | real people. Promote community engagement. | |
| Collective responsibility | High awareness of social benefits of vaccination | Provide information about collective benefits and herd immunity. Stress social benefit. | How Herd Immunity Works |
| | | Policy considerations and role of mandates to improve coverage; awareness of prerequisites for mandates and positive and negative consequences. | Policy Considerations for Mandatory COVID-19 Vaccination |
| Willingness to protect others and to eliminate infectious diseases. | High empathy for persons who are vulnerable | Use empathy with vaccine hesitant individuals. Provide visual material of other vulnerable people who benefit if people align with vaccine recommendations. | Building Trust and Empathy Around COVID-19 |
| Compliance | High awareness of positive social norms | Communicate positive social, cultural, and religious norms. | Cultural Differences in Vaccine Acceptance |
| Support for societal monitoring and sanctioning of people who are not vaccinated. | High awareness of scientific consensus | Communicate doctors' consensus on trust in vaccination. | Communicating Doctors' Consensus Increases COVID-19 Vaccinations |
| Conspiracy | Pre-bunking and low endorsement of conspiracy theories | Use psychological inoculation or pre-bunking to explain to audiences how they might be misled before misinformation is encountered. | Short Videos that Inoculate against Misinformation Online |
| Conspiracy thinking and belief in fake news related to vaccination. | Effective debunking and availability of correct information | Engage with social media to disseminate correct information and protect the public against misinformation. | How to respond to vocal vaccine deniers in public? Debunking Handbook 2020 JITSUVAX Learning Resource to Debunk Vaccine Disinformation Debunkings of Common Myths about COVID-19 A Manifesto for Science Communication as Collective Intelligence |

Links with specific examples on how to apply the interventions are provided for COVID-19 vaccination as a case example. Definitions of antecedents of vaccination readiness are adapted from[1].

compliance, and conspiracy. Table 1 summarizes the 7 C framework and the corresponding interventions, informed by learnings from the COVID-19 vaccine rollout during the last 2 years, and points to sources with advice to practitioners. Although the relative importance of the various factors in the 7 C framework may differ between vaccines, cultural contexts, and populations, we consider the interventions available for each factor to be relatively stable. Box 1 illustrates how those recommendations can be put into practice in a stylized conversation with a patient. Although we emphasize the learnings from the pandemic, our interventions are broad and can apply to many contexts in which people are hesitant about being vaccinated.

While psychological factors are the focus of this paper, we acknowledge that low vaccine uptake is complex and multifactorial, and effective solutions to address it often involves interventions that address both individual, social, cultural and structural factors.

## Confidence

Confidence refers to "trust in the safety and effectiveness of vaccinations, the health authorities, and the health officials who recommend and develop vaccines"[1]. Physicians are one of the most trusted sources of health information and even individuals with low vaccination readiness consider their health care providers (HCPs) to be the most trusted source for vaccine information. Ensuring vaccination readiness among HCPs is thus crucial

both in terms of increasing their own vaccine uptake to reduce the burden of disease, but also given their role in promoting vaccination in their communities. One way to overcome low levels of vaccination readiness is trying to understand reasons for individuals' concerns by asking open questions, using reflective listening, and building a trusting relationship during conversations[2]. The critical role of HCPs underpins the need for effective capacity building and training around such vaccine communication skills. Learning materials that provide guidance for HCPs to apply these skills in conversation are provided by WHO and other organisations (Table 1).

Trust in institutions is another component of confidence and a crucial determinant of vaccine uptake, and trustworthiness of institutions must be maintained by transparent communication. Although one might fear that disclosing negative information may increase vaccine hesitancy, studies conducted during the pandemic suggest that transparent communication sustains trust in health authorities and hinders spread of conspiracy beliefs.

## Complacency

Complacency is defined as reluctance "to get vaccinated due to low perceived risk of infectious diseases"[1]. Perceived risk of the disease is known to be a particularly important factor underpinning vaccination readiness, and this was the case for COVID-19 as well[3]. Notwithstanding public health messaging and guidelines recommending vaccination for prevention of severe

**Box 1 | Suggested communication approach when an adult is hesitant about a further COVID-19 booster vaccination. (Based on ref. [13])**

| Communication | Technique | Example |
|---|---|---|
| Ask about questions and concerns | Try to elicit top 3 concerns using open ended questions<br>Allow adequate time/listen, before addressing the questions | "Are you up to date with your recommended vaccines? If not, what is keeping you from being vaccinated?" |
| Reflect and summarise | Summarise concerns to check understanding<br>Establish rapport through non- judgement | "So if I can summarise, it sounds as though your main concern is that…" |
| Acknowledge concerns | State that expressing concerns is normal | "Having questions is very normal " |
| Share knowledge | Offer to share your knowledge Avoid over-reassurance Acknowledge all vaccines have side effects<br>Guide towards trustworthy sources of information | "Can I share what I know? Protection provided by COVID-19 vaccines wanes over time, especially for protection against severe disease or you getting sick enough to go to hospital. You need a booster dose to keep your protection up" |
| Reinforce motivation to vaccinate | Share stories about people with risk factors who got sick | "Can I tell you about a patient, who had diabetes like you and got seriously ill because he had not been vaccinated yet." |
| Discuss disease severity | Return to talk about the severity of the disease, not just the vaccines | "Diseases like COVID and flu can be more severe as we get older and in people with underlying medical conditions…" |
| Recommend vaccination | Always finish with a recommendation to vaccinate Explain where to get vaccinated/how to get it | "The best way to protect you and to ensure you can enjoy your life freely and travel is to have a vaccine now" |
| Leave the door open | If they are not ready, offer to continue the conversation later | "It seems like you are not ready to make a decision today. Maybe we can talk again in a few weeks" |

infectious diseases, experiencing no or only mild symptoms can lead people to underestimate the danger of a virus and hence the utility of the vaccine. One way to make possible serious consequences more tangible is through the communication of narratives. Those narratives must however avoid creating fear without also increasing perceived self-efficacy. For further resources on how to communicate individual case reports and still have a critical evidence-based dialogue, see Table 1.

## Constraints

Constraints in this context refer to "psychological hurdles that make vaccination difficult"[1]. Reduced access to trustworthy information has been a critical constraint for many, and various studies have shown that low vaccination rates among ethnic minorities are often not primarily due to anti-vaccination beliefs or ideology, but to a lack of transparent and accessible information. The European Centre for Disease Prevention and Control (ECDC) provides guidance on how to adapt scientific results to different cultural realities, and UNICEF provides information materials on how to make risk communication more accessible to marginalised and vulnerable groups, such as persons with disabilities, indigenous populations, refugees, or children (Table 1). In some cases stress or workload may impair people's ability to think about a vaccination decision. This can be addressed, for example, by making people aware of stress coping strategies – some useful insights are provided by the Centers for Disease Control and Prevention (CDC; Table 1).

## Calculation

Calculation refers to the "degree to which personal costs and benefits of vaccination are weighted"[1]. The actual availability and quality of information as well as the individual ability to obtain or understand health information (health literacy) can affect this factor. A qualitative study in Australia amongst vaccine-hesitant adults made 11 recommendations to address communication content, delivery, and context to increase uptake and acceptance of COVID vaccines. The recommendations include the need to communicate about vaccine safety and effectiveness (and weigh risk), to address concerns about expected side effects, highlight benefits of vaccination and discuss disease severity to counter a 'wait and see' approach and to communicate about vaccine availability.

A wide-ranging comparison of possible interventions in the UK found that messages centreing on the personal benefit accruing from COVID-19 vaccinations were highly effective and increased readiness more than, for example, information on collective benefits[4]. These results support the idea that people engage in, and are sensitive to, a personal risk calculus, although in other cultural contexts people are also sensitive to collective risks and benefits. A personal risk calculus is important as some people may believe that a vaccine is riskier to them individually due to their health condition, age, or other factors, such as fertility, pregnancy and breastfeeding. People understandably like information to be personalized to their risks and needs.

Information that takes individual differences in health literacy into account is critical to the dissemination of scientific information. Studies indicate that information that is more complex and less comprehensible is less likely to be shared[5]. Several decision aids provide an instrument for making risk calculations available to laypersons in a comprehensible form, which can contextualise and support decisions about whether to get a vaccine or oneself or one's child. One example of a decision aid was provided by the National Centre for Immunisation Research and Surveillance (NCIRS) in Australia (Table 1).

## Collective responsibility

Collective responsibility is defined as "willingness to protect others and to eliminate infectious diseases" through collective action[1]. Collective responsibility is an essential ingredient to any vaccination programme because vaccines benefit both the vaccinated and those around them, and if enough people are vaccinated, herd immunity may be achieved. A decision to get

vaccinated is therefore a prosocial decision[6], which is also reflected by correlations of vaccination readiness with the pro-social personality traits of honesty-humility and agreeableness[1].

Consistent with this reasoning, it has been demonstrated that providing information about herd immunity in the case of COVID-19 increases COVID-19 vaccination intentions, which is increased further when inducing empathy for persons who are particularly vulnerable to the disease. Guidance documents on how to build empathy and simulations to communicate herd immunity are available (Table 1). Relatedly, stressing the societal benefits of high uptake rates achieved through vaccination mandates decreased psychological reactance (i.e., anger) towards such policies.

## Compliance

Compliance is defined as "support for societal monitoring and sanctioning of people who are not vaccinated"[1]. Social and cultural norms, and religious, family, and community influence have been found to be important determinants for routine, influenza, and COVID-19 vaccination alike. Existing positive social norms could therefore be leveraged to increase vaccine uptake, such as communicating the scientific consensus on vaccination. A recent study in the Czech Republic demonstrated that communicating such consensus indeed increased COVID-19 vaccine uptake rates x. The communication materials that were used to achieve this effect are publicly available (Table 1).

To enhance uptake, many countries have introduced mandates. These were largely supported by HCPs with the caveat that they should be implemented with careful planning and consultation to avoid unintended consequences. A recent analysis has shown that mandates are quite effective overall at increasing uptake of the mandated vaccine. There are, however, other consequences to mandates such as disgruntlement that may have long-term adverse consequences[7]. Reactance to COVID-19 vaccination mandates has been observed in studies in Germany and the U.S. The longer lasting negative impact of mandates on vaccine trust and confidence as well as social polarization remains to be evaluated[8].

## Conspiracy

Endorsement of conspiracies is a strong predictor of vaccination hesitancy[9]. The continuously evolving and sometimes shifting scientific findings during the pandemic have provided fertile ground for conspiracies. Priority approval procedures for vaccines and the use of relatively new vaccine platforms have added to this perceived capriciousness of scientific knowledge, fuelled further by disinformation, undermining confidence in vaccine safety. Health care professionals may themselves also be susceptible to COVID-19 conspiracy theories.

A strong association between perceived believability of COVID-19 misinformation and low vaccination readiness has been reported in a survey spanning 40 countries[10]. In addition, randomized controlled trials reveal that exposure to COVID-19 vaccination misinformation can increase the belief in false statements (e.g., vaccination causes cancer) and decrease the intention to get vaccinated[11].

Several interventions have shown promise in combating mis-information. For example, choosing trusted messengers can increase COVID-19 vaccination readiness in target groups. Likewise, inoculation messages that explain to audiences how they might be misled *before* the misinformation is encountered have been repeatedly shown to be effective. Videos that have been proven to produce such inoculation effects are publicly available (see Table 1; the table also provides resources on how to debunk misinformation using a fact-sandwich structure in communication and how to rebut misinformation in public debates).

## Conclusions

The COVID-19 vaccines are a scientific and public health success story[12], having prevented an estimated 20 million deaths within a year of their introduction. Nonetheless, suboptimal vaccine uptake remains a challenge in many countries globally. Separate to access barriers, a considerable body of behavioural research has emerged related to the psychological factors associated with vaccine hesitancy. The 7 C theoretical framework helps to understand hesitancy towards COVID-19 as well as vaccine hesitancy more widely. This research also produced evidence-informed interventions that can help increase vaccine uptake, separate to addressing access barriers. Those interventions are summarized in Table 1 and provide a broad toolbox that can address the various drivers of hesitancy beyond the specific context of COVID-19.

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

## Acknowledgements

S.L., D.H., P.S., C.B., A.S., and A.F. acknowledge support from the European Commission (Horizon 2020 grant 964728 JITSUVAX). T.S.H. acknowledges support from a Ford Foundation Senior Fellowship (2022-2023). C.B. also acknowledges support from the Leibniz Association, #P106/2020. H.S. has received funding from Moderna for investigator-driven research and travel costs. This funding was not associated with this work. HJL's research group has received research funding from GSK, Merck and Janssen.

This funding was not associated with this work. None of the funders had a role in the preparation of the manuscript and decision to publish. The authors are responsible for the views expressed in this article and do not necessarily represent the views, decisions or policies of the institutions with which they are affiliated.

## Author contributions

S.L.: conceptualization and management of project, final draft, revision, editing; P.S.: conceptualization, lead design and construction of Table 1; K.B.H.: conceptualization, first draft, editing; M.D.: Box 1, editing; all other authors: contributing to conceptualization, provide comments on drafts, editing.

## Competing interests

C.S. is a Special Government Employee (unpaid) at the U.S. Department of Homeland Security and serves on the WHO Technical Advisory Group on Behavioral Insights and Sciences for Health (also unpaid). All other authors declare no competing interests.

## Additional information

[1]School of Psychological Science, University of Bristol, Bristol, UK. [2]Department of Psychology University of Potsdam, Potsdam, Germany. [3]School of Psychological Science, University of Western Australia, Nedlands, WA, Australia. [4]Centre for Language Studies, Radboud University Nijmegen, Nijmegen, Netherlands. [5]Health Communication, Department of Implementation Research, Bernhard-Nocht-Institute for Tropical Medicine, Hamburg, Germany. [6]Institute for Planetary Health Behaviour, University of Erfurt, Erfurt, Germany. [7]WHO Regional Office for Europe, Copenhagen, Denmark. [8]School of Population Health, University of New South Wales, Sydney, Australia. [9]Faculty of Psychology, University of Vienna, Vienna, Austria. [10]Department of Psychology, University of Copenhagen, Copenhagen, Denmark. [11]Center for Social Data Science, University of Copenhagen, Copenhagen, Denmark. [12]Harvard University, Harvard Law School, Cambridge, MA, USA. [13]Cornell University, Ithaca, NY, USA. [14]Dalhousie University, Halifax, Nova Scotia, Canada. [15]Department of Anthropology, Laval University, Quebec City, Québec, Canada. [16]Department of Behavioural Science and Health, University College London, London, UK. [17]Department of Infectious Disease Epidemiology, London School of Hygiene & Tropical Medicine, London, UK. [18]Institute for Health Metrics & Evaluation, University of Washington, Seattle, WA, USA. [19]WHO Country Office, Mykhaila Hrushevskoho St, 9B, Kyiv, Ukraine. [20]National University of Kyiv-Mohyla Academy, Kyiv, Ukraine. [21]Center for Culture-centered Approach to Research and Evaluation (CARE), Massey University, Palmerston North, New Zealand. [22]School of Medicine, Aristotle University of Thessaloniki, Thessaloniki, Greece. [23]Department of Sociology and Social Work, Tbilisi State University, Tbilisi, Georgia. [24]Institute of Clinical Medicine, University of Turku, Turku, Finland. [25]Department of Developmental Psychology and Socialization, University of Padova, Padova, Italy. [26]Institute of Behavioural Sciences, Semmelweis University, Budapest, Hungary. [27]Department of Mental Health, Heim Pal National Pediatric Institute, Budapest, Hungary. [28]Unit of Epidemiological Psychiatry and Evaluation, IRCCS Istituto Centro San Giovanni di Dio Fatebenefratelli, Brescia, Italy. [29]Epidemiology National Centre, Carlos III Health Institute, Madrid, Spain. [30]University of Murcia, Murcia, Spain. [31]Centro Nacional de Medicina Tropical, Instituto de Salud Carlos III, Madrid, Spain. [32]Centro Nacional de Epidemiología, Instituto de Salud Carlos III, Madrid, Spain. [33]Public Health Agency of Sweden, Stockholm, Sweden. [34]European University at St.Petersburg, St, Petersburg, Russia. [35]Aarhus University, Aarhus, Denmark. [36]Department of Anthropology, Vanderbilt University, Nashville, TN, USA. [37]Faculty of Medicine, University of Coimbra, Coimbra, Portugal. [38]Centre for the Evaluation of Vaccination, University of Antwerp, Antwerp, Belgium. [39]Vaccine Uptake group, Murdoch Childrens Research Institute, Flemington Road Parkville, Melbourne, Australia. [40]Department of Paediatrics, University of Melbourne, Flemington Road Parkville, Melbourne, Australia. [41]Department of General Medicine, Royal Childrens Hospital, Flemington Road Parkville, Melbourne, Australia. ✉email: stephan.lewandowsky@bristol.ac.uk

