## [Peer Review File · Communications Psychology]

This manuscript has been previously reviewed at another Nature Portfolio journal. The manuscript was considered suitable for publication without further review at Communications Psychology

25th Oct 23

Dear Steve,

Your Comment titled "Lessons from COVID-19 for behavioural and communication interventions to enhance vaccine uptake" has now been evaluated by us. Taking into account the changes you undertook in response to the earlier reviews and editorial requests, we are happy, in principle, to publish it in Communications Psychology under a Creative Commons 'CC BY' open access license without charge.

If the revised paper is in Communications Psychology format, in accessible style and of appropriate length, we shall accept it for publication immediately. I have attached a checklist to aid you in preparing your file.

* Communications Psychology uses a transparent peer review system. The previous reviewer reports (originating with another journal) and your replies will not be included.

*If you have not done so already, please alert me to any related manuscripts from your group that are under consideration or in press at other journals, or are being written up for submission to other journals (see www.nature.com/authors/editorial_policies/duplicate.html for details).

FORMATTING GUIDELINES:

You will find a complete list of formatting requirements following this link:

<https://www.nature.com/documents/commsj-style-formatting-checklist-comment.pdf>

Please use the checklist to prepare your manuscript for final submission. In the following, I also highlight some issues of particular importance.

* References

References appear as superscript Arabic numerals, in order of mention. The reference list mentions references in the numerical order in which they are mentioned in the main text. If a reference is cited more than once, the same number is used throughout the text and the reference receives a single entry in the reference list.

Only papers that have been published or accepted by a named publication should be in the reference list (preprints and citations of datasets are also permitted). Unpublished/Submitted research should not be included in the reference list; it should only be mentioned briefly and parenthetically in the main text. Note that no major arguments should rely on unpublished research.

Published conference abstracts and URLs for web sites should be cited parenthetically in the text, not in the reference list.

Footnotes are not used.

* Competing interests

Please include a "Competing interests" statement after the References. Note that we ask authors to

declare both financial and non-financial competing interests. For more details, see <https://www.nature.com/authors/policies/competing.html>. If all or some authors have no financial or non-financial competing interests, please state so: "The/ the other / authors declare no competing interests."

SUBMISSION INFORMATION:

* Your paper will be accompanied by a two-sentence editor's summary, of between 250-300 characters, when it is published on our homepage. We suggest:
The Covid pandemic has yielded new insights into psychological vaccine acceptance factors. This knowledge serves as a basis for behavioral and communication interventions that can increase vaccination readiness for other diseases.

In order to accept your paper, we require the following:

- * A cover letter describing your response to our editorial requests.
- * The final version of your text as a Word or TeX/LaTeX file, with any tables prepared using the Table menu in Word or the table environment in TeX/LaTeX and using the 'track changes' feature in Word.

At acceptance, the corresponding author will be required to complete an Open Access Licence to Publish on behalf of all authors, declare that all required third party permissions have been obtained.

Please note that your paper cannot be sent for typesetting to our production team until we have received this information; **therefore, please ensure that you have this ready when submitting the final version of your manuscript.**

ORCID

Communications Psychology is committed to improving transparency in authorship. As part of our efforts in this direction, we are now requesting that all authors identified as 'corresponding author' create and link their Open Researcher and Contributor Identifier (ORCID) with their account on the Manuscript Tracking System (MTS) prior to acceptance. ORCID helps the scientific community achieve unambiguous attribution of all scholarly contributions. For more information please visit <http://www.springernature.com/orcid>

For all corresponding authors listed on the manuscript, please follow the instructions in the link below to link your ORCID to your account on our MTS before submitting the final version of the manuscript. If you do not yet have an ORCID you will be able to create one in minutes.

IMPORTANT: All authors identified as 'corresponding author' on the manuscript must follow these

instructions. Non-corresponding authors do not have to link their ORCID's but are encouraged to do so. Please note that it will not be possible to add/modify ORCID's at proof. Thus, if they wish to have their ORCID added to the paper they must also follow the above procedure prior to acceptance.

To support ORCID's aims, we only allow a single ORCID identifier to be attached to one account. If you have any issues attaching an ORCID identifier to your MTS account, please contact the Platform Support Helpdesk.

[link redacted]

We hope to hear from you within two weeks; please let us know if the process may take longer.

Best wishes,

Marika

Marika Schiffer, PhD
Chief Editor
Communications Psychology